∂ | **Open Peer Review** | *Veterinary Microbiology* | *New-Data Letter*

# First report on the identification and characterization of mammalian orthoreovirus from sheep in China

Dengshuai Zhao,[1] Ping Li,[1,2] Yuanhang Zhang,[1] Dixi Yu,[1] Tianyu Wang,[1] Keshan Zhang[1]

**KEYWORDS**   mammalian orthoreovirus, sheep, whole genome sequencing, phylogenetic analysis

Mammalian orthoreovirus (MRV) infects various mammalian species worldwide, including humans, pigs, bats, cattle, ferrets, dogs, cats, and civets (1–9). Various reports of clinical cases of MRV in humans and other mammalian hosts have presented evidence confirming the possibility of interspecies transmission (10, 11). MRV typically presents as asymptomatic or mild respiratory and gastrointestinal infections (12). MRV is capable of inducing neurological symptoms. However, cases with neurological symptoms are less reported than those with gastrointestinal and respiratory symptoms. Infection with MRV types 2 and 3 may result in necrotizing encephalopathy and meningitis (13, 14), whereas MRV types 1 and 3 may infect the central nervous system of mice via various pathways (15, 16). Thus, MRV poses a risk of interspecies transmission and can potentially cause severe clinical symptoms.

Here, we report the first identification of MRV from sheep in China. The sheep were located in high-altitude regions and showed clinical symptoms of watery diarrhea and dehydration. We collected multiple serum and fecal samples as well as anal swabs and tissue specimens from various segments of the gastrointestinal tract, including the duodenum, jejunum, ileum, and cecum. Agarose gel electrophoresis revealed the presence of MRV in the samples. Further analysis using whole genome sequencing revealed that the samples were MRV positive. We utilized high-throughput sequencing technology to obtain the complete genomic sequence of the MRV strain (GR/2023) and determined the lengths of 10 genomic segments, along with the proteins they potentially encode (GenBank accession nos. OR902351–OR902360) as follows: *L1*: 3,864 bp (λ3: 1,287 aa), *L2*: 3,870 bp (λ2: 1,289 aa), *L3*: 3,840 bp (λ1: 1,279 aa), *M1*: 2,244 bp (μ2: 747 aa), *M2*: 2,127 bp (μ1: 708 aa), *M3*: 2,166 (μNS: 721 aa), *S1*: 1,425 bp (σ1: 474 aa and σ1s: 147 aa), *S2*: 1,323 bp (σ2: 440 aa), *S3*: 1,161 bp (σNS: 386 aa), and *S4*: 1,176 bp (σ3: 391 aa).

The σ1 protein, encoded by the *S1* segment, is crucial for cellular attachment, type-specific serum neutralization, and hemagglutinin activity. Furthermore, σ1 protein variations are used to classify MRVs into four serotypes: type 1 (MRV1) Lang, type 2 (MRV2) Jones, type 3 (MRV3) Dearing, and putative type 4 (MRV4) Ndelle (17, 18). We further compared the nucleotide sequence of GR/2023 with those of other MRVs from various hosts and serotypes, revealing a similarity of 97.15%–99.65% with MRVs from different hosts and serotypes. Among them, GR/2023 exhibited the highest nucleotide sequence similarity (99.65%) with the bat-derived MRV strain WIV3 (KT444578). A phylogenetic analysis was conducted based on the *S1* gene (Fig. 1A). The findings indicated that GR/2023 clustered within the MRV2 lineage and exhibited the closest genetic relationship with WIV3. Similarly, the pairwise genetic distance heat map, derived from the coding sequence of the *S1* gene fragment and the inferred amino acid sequence, demonstrated that GR/2023 shared the highest similarity with WIV3 (Fig.

Address correspondence to Keshan Zhang, vetzks009@163.com.

Dengshuai Zhao and Ping Li contributed equally to this article. Author order was determined by drawing straws.

The authors declare no conflict of interest.

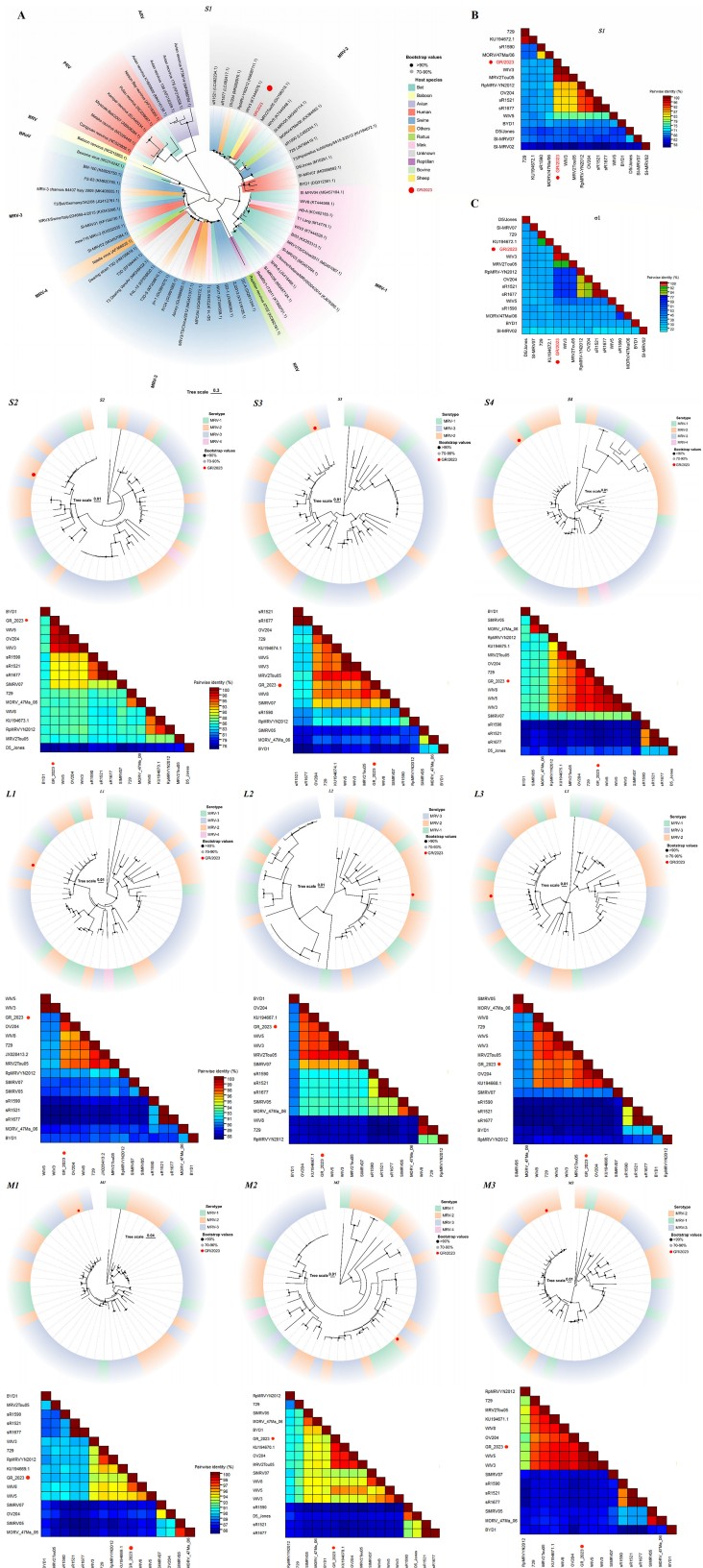

**FIG 1** Phylogenetic analysis of GR/2023. (A) Phylogenetic tree based on the *S1* genomic segment of GR/2023 and the most relevant coding sequences of other positive-sense reovirus strains in GenBank. The tree was constructed using the neighbor-joining method in MEGA software with 1,000 bootstrap

**Fig 1 (Continued)**

replicates. Different MRV lineages and virus hosts are marked with different color blocks, and GR/2023 is represented by a red circle. (B) Heatmap of the pairwise genetic distance between the coding sequence of the GR/2023 *S1* fragment and other MRV2 strains was constructed using Sequence Demarcation Tool Version 1.2 (SDTv1.2) software. The red circle represents GR/2023. (C) Paired genetic distance heatmap based on the amino acid sequence of σ1, GR/2023, and other MRV2 strains using SDTv1.2 software. The red circle represents GR/2023. *S2–S4*, *L1–L3*, and *M1–M3* phylogenetic analysis and pairwise genetic distance heatmap of GR/2023 based on the complete nucleotide coding sequences of *S2–S4*, *L1–L3*, and *M1–M3* segments. Different MRV lineages are marked with different color blocks. The red circle represents GR/2023. The phylogenetic tree was constructed in MEGA software using the neighbor-joining method. The evolutionary tree was beautified using tvBOT (19). The pairwise genetic distance heatmap was constructed using SDTv1.2 software.

1B and C). These findings suggest that GR/2023 may have undergone recombination events with WIV3 and other strains. Although phylogenetic analysis based on the *S1* gene indicated that GR/2023 belonged to MRV2, analyses based on other genes revealed lower levels of homology between GR/2023 and MRV2 (Fig. 1 *S2–S4*, *L1–L3*, and *M1–M3*). Previous analyses of nucleotide sequence similarity revealed that *L1* and *L3* exhibited the highest similarity with MRV3 strain SD-14 (98.21%) and MRV1 strain WIV2 (98.92%), derived from *Mink* and *Myotis ricketti*. Pairwise genetic distance analysis based on the amino acid sequences encoded by *L1* and *L3* also showed lower similarity with MRV2 strains compared with other gene coding sequences (Fig. 1 L1, L3). In our study, GR/2023 *L2* and *S2* showed the highest similarity with the human-derived MRV2 Tou05 strain (97.15%) and Osaka2005 strain (98.94%), respectively, providing further evidence indicating that GR/2023 originated from the recombination of WIV3 with other strains from diverse origins. This discovery is of paramount importance for further exploration of the epidemiologic and molecular characteristics of ovine-origin MRVs, as well as understanding the diversity of MRV hosts and recombination events. Additional research focusing on the development of a diagnostic kit and an effective vaccine is crucial for developing effective control and prevention strategies as well as protecting the health and productivity of sheep herds in China.

## ACKNOWLEDGMENTS

This research was funded by the National Key Research and Development Program Projects (2023YFD1801301, 2023YFD1801302) and the High-level Talent, Lingnan Scholar Research Initiation Fund (CGZ07001).

## AUTHOR AFFILIATIONS

[1]Guangdong Provincial Key Laboratory of Animal Molecular Design and Precise Breeding, College of Animal Science and Technology, Foshan University, Foshan, Guangdong Province, China
[2]College of Veterinary Medicine, Henan Agricultural University, Zhengzhou, China

## AUTHOR ORCIDs

Dengshuai Zhao http://orcid.org/0009-0008-2613-0598

## AUTHOR CONTRIBUTIONS

Dengshuai Zhao, Data curation, Methodology, Software, Writing – original draft, Writing – review and editing | Ping Li, Data curation, Methodology, Writing – original draft, Writing – review and editing | Yuanhang Zhang, Data curation, Formal analysis, Software | Dixi Yu, Data curation, Formal analysis | Tianyu Wang, Data curation, Software | Keshan Zhang, Formal analysis, Funding acquisition, Writing – original draft, Writing – review and editing

## DATA AVAILABILITY

Sequences have been submitted to GenBank under accession numbers OR902351–OR902360.

## ADDITIONAL FILES

The following material is available online.

### Open Peer Review

**PEER REVIEW HISTORY (review-history.pdf).** An accounting of the reviewer comments and feedback.

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
