## [Reviewer comments · Microbiology Spectrum]

Microbiology Spectrum

First Report on the Identification and Characterization of Mammalian Orthoreovirus from Sheep in China

Dengshuai zhao, Ping Li, Yuan-hang Zhang, Dixi Yu, TianYu Wang, and Keshan Zhang

Corresponding Author(s): Dengshuai zhao, Foshan University School of Life Science and Engineering

Review Timeline:

Submission Date:	May 26, 2024
Editorial Decision:	June 29, 2024
Revision Received:	July 8, 2024
Accepted:	July 11, 2024

Editor: Holly Ramage

Reviewer(s): The reviewers have opted to remain anonymous.

Transaction Report:

DOI: <https://doi.org/10.1128/spectrum.00847-24>

Re: Spectrum00847-24 (First Report on the Identification and Characterization of Mammalian Orthoreovirus from Sheep in China)

Dear Dr. Dengshuai zhao:

Thank you for the privilege of reviewing your work. Below you will find my comments, instructions from the Spectrum editorial office, and the reviewer comments.

Revision Guidelines

Sincerely,
Holly Ramage
Editor
Microbiology Spectrum

Reviewer #1 (Comments for the Author):

Mammalian orthoreovirus (MRV) infects many mammalian species including humans, bats, and domestic animals. Although infections in humans are typically mild, in animals, especially livestock in agriculture, this virus can cause economic losses. The research provides a comprehensive analysis of the genetic evolution of MRV from sheep (GR/2023), encompassing all its genes. The authors' results indicate that GR/2023 clusters within the MRV2 lineage and is most closely related to WIV3. Furthermore,

based on the nucleotide and amino acid sequences of the S1 gene, the results show that GR/2023 shares the highest similarity with WIV3. The authors speculate that GR/2023 may have originated from an MRV sourced from bats (WIV3). The authors also conducted genetic evolution analysis and homology comparisons of other genes of MRV. It was found that other genes do not exhibit high similarity to type 2 MRV, suggesting that GR/2023 may have undergone recombination. Currently, there are no reported articles on the complete genome sequence analysis of MRV from sheep. Overall, this study provides interesting results that enhance our understanding of molecular characteristics of MRVs, which may be of potential interest to Microbiology Spectrum readers.

There are a few points that need to be addressed before considering this paper suitable for publication. Please see below for the detailed points:

Please consider incorporating a discussion on the latest advancements in research on MRV, especially focusing on relevant reports and recent discoveries within China.

I suggest presenting the nucleotide and amino acid homology analysis of GR/2023 with other type 2 MRVs in the form of a table. The resolution of the figures needs to be enhanced, and necessary annotations should be added to facilitate reader comprehension.

Some minor points:

There are errors in the use of punctuation marks in both Chinese and English in the manuscript. I suggest to revise the entire manuscript, for example, Lines 28, 37, and 47.

Reviewer #2 (Comments for the Author):

The manuscript by Zhao et al. presents the first complete genome of mammalian orthoreovirus from sheep in China. The authors suggested that potential recombination events involving the mammalian orthoreovirus strain with bat-derived strain WIV3 and other strains. It would be helpful to briefly mention the significance of mammalian orthoreovirus in veterinary and public health, particularly in relation to sheep farming in high-altitude regions.

Addressing the following minor revisions will enhance the clarity and impact of the study.

1. The manuscript is generally clear but could benefit from minor language refinements for readability and precision. I think author should get this manuscript polished by some native English-speaking scholars.
2. The figure legend contains unnecessary repetitions. e.g. Sequence Demarcation Tool Version 1.2 software.
3. Line 12; replace "however" with "However"
4. Line 24; delete the word, "named"
5. Line 34; replace "different" with "various"
6. Line 12; replace "We also performed a phylogenetic analysis based on the S1 gene" with "A phylogenetic analysis was conducted based on the S1 gene"

Responses to reviewers' comments on manuscript no. Spectrum00847-24

Dear Reviewer and Editor:

We appreciate all of your comments and suggestions which have led us to generating a stronger manuscript. We believe we have responded to every reviewer comment as detailed below. The original reviewers' comments are shown in black and the authors' responses are shown in blue.

Editor comments:

Answer: Thank you for your comments. We have answered reviewer's comments one by one.

Reviewer #1 (Comments for the Author):

Mammalian orthoreovirus (MRV) infects many mammalian species including humans, bats, and domestic animals. Although infections in humans are typically mild, in animals, especially livestock in agriculture, this virus can cause economic losses. The research provides a comprehensive analysis of the genetic evolution of MRV from sheep (GR/2023), encompassing all its genes. The authors' results indicate that GR/2023 clusters within the MRV2 lineage and is most closely related to WIV3. Furthermore, based on the nucleotide and amino acid sequences of the S1 gene, the results show that GR/2023 shares the highest similarity with WIV3. The authors speculate that GR/2023 may have originated from an MRV sourced from bats (WIV3). The authors also conducted genetic evolution analysis and homology comparisons of other genes of MRV. It was found that other genes do not exhibit high similarity to type 2 MRV, suggesting that GR/2023 may have undergone recombination. Currently, there are no reported articles on the complete genome sequence analysis of MRV from sheep. Overall, this study provides interesting results that enhance our understanding of molecular characteristics of MRVs, which may be of potential interest to Microbiology Spectrum readers.

Answer: We thanks the reviewer for their kind and careful evaluation of the paper. And we also thanks for your positive comments for our manuscript.

There are a few points that need to be addressed before considering this paper suitable for publication. Please see below for the detailed points:

Please consider incorporating a discussion on the latest advancements in research on MRV, especially focusing on relevant reports and recent discoveries within China.

Answer: Thank you for your suggestion. Currently, the complete genome of MRV in sheep has not been reported. We will delve deeper into the discussion and analysis of GR/2023 in our subsequent work.

I suggest presenting the nucleotide and amino acid homology analysis of GR/2023 with other type 2 MRVs in the form of a table.

Answer: Thank you for your suggestion. We noticed that the format requirements of the *Microbiology Spectrum* Journal stipulate that Letters should include only one figure or table. Therefore, we regret that we did not present specific homology data in tabular form. We will consider adding the table as a supplementary file. Additionally, we believe the revised figure captions adequately explain the nucleotide and amino acid homology.

The resolution of the figures needs to be enhanced, and necessary annotations should be added to facilitate reader comprehension.

Answer: Thank you for your comment. We have enhanced the resolution of the figures and added necessary annotations to facilitate better reader comprehension. We believe these improvements will enhance the clarity and accuracy of the information conveyed.

Some minor points:

There are errors in the use of punctuation marks in both Chinese and English in the manuscript. I suggest to revise the entire manuscript, for example, Lines 28, 37, and 47.

Answer: Thanks for your comment. We have carefully reviewed and revised the entire manuscript to correct errors in the use of punctuation marks. Specific corrections have been made on lines 28, 37, and 47, as well as throughout the manuscript. (Lines 26, 35, and 45)

Reviewer #2 (Comments for the Author):

The manuscript by Zhao et al. presents the first complete genome of mammalian orthoreovirus from sheep in China. The authors suggested that potential recombination events involving the mammalian orthoreovirus strain with bat-derived strain WIV3 and other strains. It would be helpful to briefly mention the significance of mammalian orthoreovirus in veterinary and public health, particularly in relation to sheep farming in high-altitude regions.

Answer: Thank you for your comment. We have revised the manuscript in accordance with the comments and marked all the amends on our revised manuscript.

Addressing the following minor revisions will enhance the clarity and impact of the study.

1.The manuscript is generally clear but could benefit from minor language refinements for readability and precision. I think author should get this manuscript polished by some native English-speaking scholars.

Answer: Thank you for your comment. We have ensured the manuscript undergoes further polishing by native English-speaking scholars to enhance its clarity, readability, and precision.

2.The figure legend contains unnecessary repetitions. e.g. Sequence Demarcation Tool Version 1.2 software.

Answer: Thank you for your feedback. We have revised the figure legends to remove unnecessary repetitions.

3.Line 12; replace "however" with "However"

Answer: Thanks for your comment. “however” has been changed to “However” (Line 12).

4.Line 24; delete the word, "named"

Answer: Thanks for your comment. “named” has been deleted (Line 24).

5.Line 34; replace "different" with "various"

Answer: Thanks for your comment. “different” has been changed to “various” (Line 34).

6.Line 12; replace "We also performed a phylogenetic analysis based on the S1 gene" with "A phylogenetic analysis was conducted based on the S1 gene"

Answer: Thanks for your comment. “We also performed a phylogenetic analysis based on the S1 gene” has been changed to “A phylogenetic analysis was conducted based on the S1 gene” (Line 37).

We thank all reviewers for their helpful and instructive comments. We hope that the revised manuscript will be acceptable for publication in *Microbiology Spectrum*.

Sincerely

Dengshuai Zhao

Re: Spectrum00847-24R1 (First Report on the Identification and Characterization of Mammalian Orthoreovirus from Sheep in China)

Dear Dr. Dengshuai zhao:

Thank you for submitting a revised version of your manuscript. I am pleased to let you know that your manuscript has been accepted, and I am forwarding it to the ASM production staff for publication. Your paper will first be checked to make sure all elements meet the technical requirements. ASM staff will contact you if anything needs to be revised before copyediting and production can begin. Otherwise, you will be notified when your proofs are ready to be viewed.

Sincerely,
Holly Ramage
Editor
Microbiology Spectrum